# Understanding Intervertebral Disc Degeneration: Background Factors and the Role of Initial Injury

**DOI:** 10.3390/biomedicines11102714

**Published:** 2023-10-06

**Authors:** Anna E. Karchevskaya, Yuri M. Poluektov, Vasiliy A. Korolishin

**Affiliations:** 1Institute of Higher Nervous Activity and Neurophysiology, Russian Academy of Sciences, 5A Butlerova Str., 117485 Moscow, Russia; ankar1998@yandex.ru; 2Medical Faculty, I.M. Sechenov First Moscow State Medical University of the Ministry of Health of the Russian Federation (Sechenov University), 8-2 Trubetskaya Str., 119991 Moscow, Russia; 3Engelhardt Institute of Molecular Biology, Russian Academy of Sciences, Vavilov Str. 32, 119334 Moscow, Russia; 4Department of Spinal Surgery, Burdenko Neurosurgical Institute, 4th-Tverskaya-Yamskaya Str. 16, 125047 Moscow, Russia; 5Russian Medical Academy of Postgraduate Education Studies, 2/1 Barrikadnaya Str., Building 1, 125993 Moscow, Russia; vkorolishin@mail.ru

**Keywords:** intervertebral disc degeneration, genetic factors, pathogenesis, inflammation, autoimmune

## Abstract

The etiology of intervertebral disc degeneration (IVDD) is complex and multifactorial, and it is still not fully understood. A better understanding of the pathogenesis of IVDD will help to improve treatment regimens and avoid unnecessary surgical aggression. In order to summarize recent research data on IVDD pathogenesis, including genetic and immune factors, a literature review was conducted. The pathogenesis of IVDD is a complex multifactorial process without an evident starting point. There are extensive data on the role of the different genetic factors affecting the course of the disease, such as mutations in structural proteins and enzymes involved in the immune response. However, these factors alone are not sufficient for the development of the disease. Nevertheless, like mechanical damage, they can also be considered risk factors for IVDD. In conclusion, currently, there is no consensus on a single concept for the pathogenesis of IVDD. We consider the intervertebral disc autoimmune damage hypothesis to be the most promising hypothesis for clinicians, because it can be extrapolated to all populations and does not counteract other factors. The genetic factors currently known do not allow for building effective predictive models; however, they can be used to stratify the risks of individual populations.

## 1. Introduction

Back pain is a leading cause of healthcare visits worldwide and a major contributor to temporary disability among the adult population. Degenerative disc diseases (DDDs) are most prevalent among men over 40 years of age [1,2]. Existing conservative treatment methods often fail to address the issues of discogenic pain syndrome in the long-term, leading to the necessity of surgical intervention.

IVDD (intervertebral disc degeneration) is a multifactorial condition, the pathogenesis of which remains incompletely understood. Numerous studies have focused on neuroimaging assessment of risk factors and predictors of IVDD development, genetic predisposition research, as well as clinical investigations aimed at identifying and analyzing conditions associated with disc degeneration. Despite the wealth of available data, there is still no unified concept of IVDD pathogenesis, and the role of various factors remains controversial or uncertain in the pathophysiological cascade of events [1,3,4].

The objective of this study is to explore the possibility of considering the intervertebral disc as an immune-privileged organ based on current global research in the fields of physiology and genetics, as well as the analysis of embryological and physiological development.

## 2. Embryonic Development and Maturation of IVD

During embryogenesis, when the notochord is formed a portion of its cells gives rise to the nucleus pulposus (NP). The separation of the notochord occurs through the involvement of connective tissue, which forms a fibrous ring around the future NP. The nucleus pulposus contains several types of cells, including large vacuolated notochordal cells and chondrocyte-like cells called nucleopulposocytes. Nucleopulposocytes are small spherical cells resembling joint chondrocytes, but they differ from the latter by expressing specific markers (OVOS2, CA12, CD24, HIF-α, and cytokeratin 8, 18 and 19). Nucleopulposocytes synthesize components of the extracellular matrix (aggrecan, type II collagen, hyaluronic acid). In the NP, notochordal cells and nucleopulposocytes exert paracrine effects on each other: signals from notochordal cells stimulate nucleopulposocytes to synthesize ECM and TGF-β1, which in turn triggers CTGF synthesis by notochordal cells, further activating nucleopulposocytes. TGF-β and Shh limit enzymatic degradation of the extracellular matrix by inhibiting matrix metalloproteinases (MMPs) through tissue inhibitors of metalloproteinase (TIMPs). This feedback loop helps maintain a balance between catabolic and anabolic reactions in the pulpy nucleus. Additionally, the Shh signaling pathway influences the proliferation and anabolic activity of annulus fibrosus cells and endplate chondrocytes [5].

## 3. Normal Anatomy of IVD

The intervertebral disc is composed of three components: the nucleus pulposus (NP), the annulus fibrosus (AF), and the cartilaginous endplates (CEP).

### 3.1. Nucleus Pulposus

The NP is the central part of the intervertebral disc and develops from the notochord. It contains a significant amount of proteoglycans, predominantly type II collagen fibers arranged in a random manner, aggregates of aggrecan bound to chondroitin and keratin sulfate, and radially oriented elastic fibers. This structural composition allows the intervertebral disc to absorb shock and withstand significant mechanical loads [6,7].

### 3.2. Annulus Fibrosus

The annulus fibrosus (AF) is a connective tissue that surrounds the nucleus pulposus. It is primarily composed of type I collagen fibers. Type I collagen is a heterotrimeric protein consisting of two identical α-chains and a third, different chain. The genes encoding these chains, COL1A1 and COL1A2, are present in both the nucleus pulposus and annulus fibrosus. The AF tissue contains fibroblast-like cells with an approximate density of 9000 cells/mm^3^. The collagen fibers are arranged at an angle of 30° to each other and form 15–25 concentric lamellae. The collagen fibers on the outer side are more organized compared to those facing the nucleus pulposus. The AF limits the degree of deformation of the nucleus pulposus and acts as a barrier between the nucleus pulposus and the immune system of the body [5,8].

The outer layer of the AF is innervated by branches of the sinuvertebral nerve (dorsal and dorsolateral parts of the AF) and numerous gray communicating branches of the sympathetic trunk (lateral and ventral parts of the AF). The innervation of the AF is provided by both peptide-containing and non-peptide-containing nociceptors as well as larger fibers forming proprioceptors. Blood vessels are found only in the outer layers of the annulus fibrosus. In fetuses and infants, blood capillaries penetrate the channels of the cartilaginous endplate and form loops in the space between the bone and cartilage. With age, these capillaries disappear, creating “weak spots” that can later lead to the development of subchondral plate sclerosis or Schmorl’s herniation [6,7,9,10].

### 3.3. Endplate

The cartilaginous endplates (CEPs) are thin layers of cartilage, approximately 0.5–1 mm thick. They contain aggrecan and collagen, with embedded chondrocytes. The CEPs contain both perivascular and non-vascularized nerve fibers. During intervertebral disc degeneration, an increase in innervation is observed in the CEPs [8,10].

The blood supply to the intervertebral disc is provided by two capillary plexuses that are extensions of the segmental artery. Branches of these arteries penetrate the vertebral body and branch out, providing nutrition to the nucleus pulposus through diffusion across the cartilaginous endplates. They also approach the outer layers of the annulus fibrosus [11,12].

## 4. Normal Physiology of IVD

The intervertebral disc (IVD) plays a crucial biomechanical role in transmitting compressive loads between vertebral bodies, providing mobility and maintaining spinal stability. The nucleus pulposus (NP) exhibits a high elasticity and can absorb compressive loads by altering osmotic pressure, which is attributed to the unique biochemical composition of the extracellular matrix (ECM). Nucleopulpocytes secrete TGF-β and Shh to limit ECM catabolism through TIMPs, thereby maintaining a balance [5,13,14,15].

The annulus fibrosus (AF) secretes exosomes and Sema3A proteins that prevent vascular and nerve ingrowth into the IVD tissue. It also maintains osmotic pressure generated by the NP in response to loading and provides tensile strength to the IVD [5,13,14,15].

Cells in undamaged cartilaginous endplates can produce exosomes that inhibit NP cell apoptosis, enhance collagen II synthesis, and inhibit the release of inflammatory factors such as IL-6 and IL-1β [16].

Since the disc is an avascular structure, nutrient supply to IVD cells occurs through diffusion from capillaries through pores in the subchondral plate, then through the cartilaginous endplates and into the extracellular matrix. Therefore, NP cells must adapt to a hypoxic environment. This adaptation is partly explained by the constitutive expression of hypoxia-inducible factor-alpha (HIF-α) in NP cells regardless of oxygen levels [5,17].

## 5. IVD as an Organ of Immune Privilege

During embryogenesis, the intervertebral disc (IVD) becomes isolated from the body’s immune system and is protected by the hematopoietic-nucleus pulposus (NP) barrier formed by the cartilaginous endplates (CEPs) and annulus fibrosus (AF). Cells in the nucleus pulposus and AF suppress the transmission of vascular endothelial growth factor (VEGF) signals and secrete exosomes that prevent blood vessel ingrowth into the NP. This isolation prevents antigens of the nucleus pulposus from entering the lymphoid system, where lymphocyte maturation occurs and immunological tolerance to nucleus pulposus antigens is not developed. Additionally, the Fas–Fas Ligand system secreted by nucleus pulposus cells serves as an additional protective factor against the immune system. Interaction between Fas Ligand and the Fas receptor triggers apoptosis, thereby protecting nucleus pulposus cells from cytotoxic T lymphocytes and NK cells and eliminating ingrown blood vessels in the disc [13,18,19,20,21,22,23,24].

Thus, the absence of blood vessels in the intervertebral disc as well as the functional characteristics of its cells and their ability to express Fas Ligand may serve as indicators of the “immune privilege” structure.

## 6. Pathophysiology of IVDD

### 6.1. Inflammation

The result of the inflammatory reaction in the intervertebral disc is the degeneration of all structures of the disc.

Degenerative changes in the intervertebral disc progress through the loss of proteoglycans, decreased oxygen concentration in the extracellular space, the accumulation of free radicals, crystals (calcium pyrophosphate dihydrate, cuboidal microstructures (magnesium bellucite), and hydroxyapatite), decreased pH, and increased activity of aberrant proteolytic enzymes. Changes in the composition of the extracellular matrix and pH lead to increased permeability of the blood–NP barrier and increased damage to the annulus fibrosus (AF) and nucleus pulposus (NP) [5,25,26].

Phagocytosis of crystals by intervertebral disc cells present in joints and periarticular tissues can activate the pyrin domain of the NOD-like receptor family, the “NLRP3 inflammasome”. This protein complex regulates the activity of caspase-1 and the synthesis of interleukin IL-1β [25].

Oxidative stress is intensified due to the imbalance between the processes of formation and absorption of reactive oxygen species (ROS), which is maintained by non-enzymatic and enzymatic antioxidants including glutathione (GSH), superoxide dismutase (SOD), catalase (CAT), glutathione peroxidase, ascorbic acid (vitamin C), α-tocopherol (vitamin E), and carotenoids. In case of a predominance of the formation processes over absorption, ROS activates NF-κB, MAPK, lipid pathways (phospholipases, protein kinase C (PKC), and PI3K/Akt), thereby enhancing catabolic reactions in the extracellular matrix. Accumulation of ROS slows down cell division, and telomerase activity is stopped by activating the MAPK signaling pathway, causing the cell cycle to arrest in the G0/1 phase [27].

These pathological processes are intensified by excessive mechanical loads on the intervertebral disc (IVD), which can lead to protrusion or extrusion. At the micro level, gene expression changes occur, causing cells to secrete a range of pro-inflammatory mediators. It has been shown that the longer mechanical load is sustained, the more cells begin to die. Hui Xu demonstrated a dependence of TNF-α expression on the degree and duration of stretching [28].

During degenerative changes in the nucleus pulposus (NP) cells, they begin to express a large number of inflammatory mediators such as IL-1, IL-6, IL-9, IL-12, IL-17, TNF-α, and IFN-γ, as well as the anti-inflammatory IL-38. Increased expression of IL-1 by damaged disc cells disrupts the extracellular matrix environment and leads to increased expression of matrix-degrading enzymes, particularly collagen II and aggrecan. The expression of matrix metalloproteinases (MMPs) 1, 3, 7, 9, 10, and 13, as well as ADAM (a disintegrin and metalloproteinase) proteases containing the thrombospondin motif (ADAMTS 1, 4, 5, 9, and 15), is increased. Experiments have shown that treatment of human IVD cells with TNF-α or rat NP cells with IL-1β significantly increases the expression levels of ADAMTS-4 and ADAMTS-5 and activates MAPK and NF-κB pathways [29,30,31,32].

TNF-α significantly increases SA-β-Gal expression by activating the MAPK/ERK pathway, thereby increasing the expression of aging markers (p53 and p16) and inducing cell cycle arrest in the G0/1 phase. Abnormal loads increase the number of SA-β-Gal-positive cells and the expression of aging-related genes, including p16, p27, RB, PTEN, p27KIP, p19ARF, and RAGE. It has been shown that TNF-α is abundant in NP tissues of fetuses/infants and elderly adults, while its expression is weak in adults and young adults. In childhood, cytokines may have a positive impact on the proliferative capacity of NP cells and be associated with active tissue remodeling. However, in aging, the elevated level of TNF-α indicates altered gene expression in NP cells and a predominance of degenerative processes. As degeneration of the NP progresses, telomere length gradually shortens and telomerase activity decreases, contributing to cellular aging and eventual cell death [33,34,35].

One way to protect NP cells from aging is the expression of SIRT proteins, which belong to the class III NAD+-dependent histone deacetylases. Several studies have shown that higher levels of SIRT1 lead to greater cell proliferation and a reduced apoptosis rate in NP cells. With age and increased oxidative stress, the expression of SIRT proteins decreases. SIRT1 inactivates p53, inhibits the NF-κB signaling pathway, and enhances resistance to oxidative stress by promoting autophagy. Miyazaki demonstrated that the introduction of recombinant human SIRT1 induced enhanced autophagy in cultured human NP cells and reduced mitochondrial apoptosis caused by nutrient deficiency [36].

During IVD damage, NP cells shift collagen synthesis from type II to type I, resulting in decreased disc elasticity. The synthesis of aggrecan, which normally binds to hyaluronic acid to form large aggregates, allowing tissues to withstand compressive loads, is also reduced. Ultimately, this leads to matrix dehydration in the NP. Type X collagen appears in degenerative discs and is associated with chondrocyte clusters and fissure formation in the NP. Decreased levels of aggrecan and collagen disrupt the structure of the annulus fibrosus (AF), dehydrate and fibrose the NP, and lead to calcification of cartilaginous endplates. NP dehydration disrupts cushioning, further damages NP cells, and alters hydrostatic pressure and pH, causing the remaining cells to secrete various growth factors and cytokines. This leads to macrophage infiltration in the damaged area and may initiate processes of vascularization and innervation [29,37].

### 6.2. Neoangiogenesis in IVDD Pathogenesis

Increased synthesis of IL-6, IL-8, and prostaglandin E2 (PGE2) by nucleus pulposus cells leads to the expression of nerve growth factor (NGF). In combination with increased expression of vascular endothelial growth factor (VEGF) in degenerated nucleus pulposus (NP), this cascade of reactions leads to abnormal nerve ingrowth and vascularization in previously avascular and non-innervated tissue, which may explain the development of discogenic pain. TNF-α plays a significant role in the onset of discogenic pain as it can lead to irritation of nerve roots [15,26,31,38].

In pathological conditions of intervertebral disc degeneration (IVDD), the expression of tissue inhibitor of metalloproteinase 3 (TIMP3) decreases in NP cells. TIMP3 is an inhibitor of angiogenesis, which suppresses vascular ingrowth into cells by preventing the binding of VEGF to VEGFR-2. The loss of TIMP3 expression induces inflammation, matrix degradation, and vascular ingrowth in the nucleus pulposus, representing a new mechanism of intervertebral disc degeneration [39].

The induction of vascularization and increased synthesis of chemokines by NP cells, particularly CCL2, 3, 4, 5, 7, and 13, CXCL10, and IL-8, stimulate the migration of immune cells that infiltrate the NP and produce IL-1β and TNF-α. These cytokines, in turn, enhance the synthesis of matrix metalloproteinases (MMP) and ADAMTS (a disintegrin and metalloproteinase with thrombospondin motifs) [5].

RANTES (CCL5), a chemotactic factor for monocytes, macrophages, and T-cells, was found to be elevated in discs of patients with more severe degeneration, regardless of the presence of herniation. This expression was further enhanced in response to TNF-α, and both basal and induced expression of RANTES were observed more in annulus fibrosus (AF) cells than NP cells, indicating an important role of the fibrous ring in the further development of disc damage [40].

Apart from pro-inflammatory cytokines, NP cells begin to secrete IL-38. In in vitro models, IL-38 significantly reduced TNF-α-induced expression of IL-1β, IL-6, COX-2, MMP-13, and ADAMTS-5. It also attenuated TNF-α-induced reduction in the synthesis of type II collagen and aggrecan and inhibited the NF-κB signaling pathway [41].

These findings suggest that various inflammatory mediators and signaling pathways play a role in intervertebral disc degeneration and discogenic pain, highlighting potential therapeutic targets for managing these conditions.

When the integrity of the intervertebral disc (IVD) is compromised, there is a decrease in the expression of Fas ligand (FasL) by nucleus pulposus cells due to changes in gene expression. As a result, infiltrating immune cells in the disc are not subjected to apoptosis, and macrophages begin to secrete various growth factors and an increased amount of cytokines, further enhancing the process of angiogenesis. Meanwhile, the cells themselves start to secrete soluble Fas (sFas), which, upon interaction with corresponding receptors on the membrane of NP cells, triggers their death [38,42,43].

The expression of FBXO6 (F-Box Protein 6) protein by nucleus pulposus (NP) cells is reduced in damaged discs. FBXO6 has the ability to inhibit stress-induced endoplasmic reticulum apoptosis. Additionally, the synthesis of FBXO6 is inhibited by IL-1β and miR-133a-5p, whose expression is enhanced during inflammation and activation of the NF-κB signaling pathway [44]. Activation of NF-κB, induced by TNF-α, enhances the secretion of periostin, which accelerates the aging process of NP cells through PIEZO-1. The NF-κB signaling pathway also induces the expression of MMP-1, MMP-3, MMP-9, MMP-13, and ADAMTS-5, and enhances the immune response by increasing the expression of chemokines CCL3 and CCL4 [33,45].

PIEZO1, in turn, induces complex molecular signals that involve the increased influx of calcium ions into the cell, activation of NF-κB, caspase-3, caspase-9, and extracellular signal-regulated kinases (ERK) 5 and ERK 1/2. In an experiment conducted by Yi Sun, it was shown that the greater the tissue stretching in the intervertebral disc, the stronger the expression of Piezo1 and NLRP3 mRNA and protein [46].

The accumulation of lactate, resulting from impaired nutrient diffusion due to calcification of the cartilaginous endplate (CEP) and increased anaerobic glycolysis, increases the expression of acid-sensing ion channels (ASIC). In an experimental model of degenerative changes in the intervertebral disc of rats, it was shown that the expression of ASIC1, ASIC2, and ASIC3 is significantly increased in degenerative NP cells. ASIC1a regulates apoptosis of CEP chondrocytes through Ca^2+^ and increases MMP activity through the NF-κB signaling pathway. Levels of ASIC1a and ASIC3 were significantly higher in degenerative disc tissue samples compared to normal tissue. Lactate stimulated the expression of ASIC1a/ASIC3 and activation of the NLRP3 inflammasome in NP cells, and NLRP3 then triggered pyroptosis [47,48].

These studies shed light on the molecular mechanisms involved in intervertebral disc degeneration and provide insights into potential therapeutic targets for managing this condition.

### 6.3. Neoneurogenesis and Microglial Role in the Pathophysiology of IVDD

In addition to NP cell damage, the functioning of AF cells is disrupted, which, under pathological conditions, begin to secrete specific AF exosomes. These exosomes stimulate the migration of endothelial cells to the damaged disc area and increase the expression of IL-6, TNF-α, MMP-3, MMP-13, and VEGF to “eliminate” the damaged structures. This response aims to eliminate damaged structures and metabolic waste. If this process is not completed sufficiently, inflammation takes on chronic characteristics. The expression of Sema3A protein, a member of the semaphorin class 3 family that inhibits nerve ingrowth, is reduced as the degenerative process intensifies in patients with discogenic pain. This leads to increased neoneurogenesis in the intervertebral disc [13,15,16].

Intervertebral disc degeneration (IDD) is a pathological process typically characterized as a cell-mediated response to progressive structural degeneration. The molecular mechanisms underlying IDD and low back pain (LBP) are still poorly understood, but several studies report that IDD and LBP are apparently mediated by abnormal production of pro-inflammatory cytokines, chemokines, and neurotrophins by various cell types. Hyperproduction of these mediators creates a local inflammatory microenvironment within the disc tissue and peridiscal space (especially in the case of disc herniation), exacerbating the overall degenerative condition, initiating a cascade of degenerative events such as cell apoptosis and disruption of the extracellular matrix (ECM) of the disc, and ultimately leading to pain. These pro-inflammatory factors involved in the degenerative process are produced and secreted not only by resident disc cells but also by circulating immune cells that can infiltrate the disc tissue, as well as spinal neuroglial cells. Specifically, several studies have reported the presence of inflammatory-like cells and immune cells, such as CD68+ macrophages, neutrophils, and T lymphocytes (CD4+, CD8+), infiltrating degenerated human discs in response to chemotactic stimuli generated during degeneration. Furthermore, pro-inflammatory stimuli in IDD apparently induce activation and proliferation of neuroglial cells, such as microglia and astrocytes, in the spinal dorsal horns. These findings suggest that immune regulatory cells, whether infiltrating the disc tissue or resident in the spinal cord, play a crucial role in neuroinflammation during IDD and pain generation. Therefore, a better understanding of the involvement of immune cells in IDD and LBP promises to fully comprehend the degenerative processes and identify molecular targets for potential new biological treatment approaches.

Navone, in their study on disc cell cultures, demonstrated that mechanical loading and nutrient deprivation in the matrix led to high production of IL-8, NGF, IFN-g, and IL-17. Increased mechanical loading and IL-8, NGF, IFN-g, and IL-17 contributed to microglial activation and migration to the pressure zone. This, in turn, intensified the inflammatory process in the disc tissues and the surrounding space [49].

### 6.4. Metabolic Syndrome and Diabetis Mellitus

Comorbid conditions, such as diabetes, also play an important role in the development of degenerative disc damage. Hyperosmolarity leads to DNA damage in cells through the activation of the ATM/p53/p21WAF1 pathway. This results in hypophosphorylation of the retinoblastoma protein (pRb) and cell cycle arrest in the G1 phase. Oxidative stress, which occurs as a result of diabetes, poses a risk to disc tissues, potentially causing or accelerating inflammation. The experimental model by Tsai et al. demonstrated that in rats with type I diabetes, the expression of MMP-2 is increased when exposed to advanced glycation end products (AGEs) compared to rats without diabetes [40,50].

### 6.5. Autoimmune Response in IVDD

When the intervertebral disc (IVD) is damaged, such as in protrusion/extrusion, once NP cells exit the AF and come into contact with the human immune system, the system recognizes them as foreign bodies [18,19]. Autologous nucleus pulposus primes T cells to develop into interleukin-4-producing effector cells. Autoimmune reactions occur with the activation of B cells, producing autoantibodies and cytotoxic T lymphocytes. Geiss et al. found a high concentration of activated T cells (CD4+ and CD8+), as well as activated B cells expressing immunoglobulin kappa (Ig-kappa), in porcine NP cell cultures [18].

Capossela et al. discovered antibodies (IgG) against collagen II, aggrecan, and collagen V in the degenerated IVDs of the caudal spine of the Atlantic stingray, as well as in degenerated human NP and AF cell cultures, which persisted even after cell culture “cleansing” [51,52,53].

Monchaux found that macrophages are present in IVD tissues during extrusion in dogs with disc herniation [20].

Reduced expression of FasL weakens the ability of NP cells to defend against T lymphocytes. Damaged NP cells begin to secrete pro-inflammatory agents that attract macrophages and other immune cells. These cells, in turn, produce matrix metalloproteinases that degrade the damaged part of the disc.

Silva discovered that human macrophages can elicit a powerful pro-inflammatory response during IVD degeneration and hinder the restoration of the cartilaginous endplate by suppressing the expression of aggrecan and collagen II genes in the presence of IL-1β [20,54].

IL-21 and IL-17 play a significant role in the development of autoimmune processes. Huawei Xue showed that patients with disc herniation have elevated levels of IL-21 and IL-17 in their serum. IL-21 controls the functional activity of effector T helper cells and the differentiation of Th17 cells, as well as promoting B cell differentiation [55,56].

On the other hand, the autoimmune process may be explained by the fact that NP cells produce carbohydrate-based extracellular matrix substances that can mimic the antigenic structure of pathological microbes, potentially triggering an immune response to the NP [57].

## 7. Role of Genetic Factors in the Development of IVDD

Currently, there is a significant amount of research dedicated to studying the role of genetic factors in the development of IVDD. However, direct evidence that specific genetic factors play a substantial role in the disease development is still lacking. It is more likely that gene polymorphisms of structural proteins and certain enzymes may act as predisposing or aggravating factors in the development of IVDD.

### 7.1. MiRNA as an Integral Indicator of IVDD Development

MicroRNAs (miRNAs) are small non-coding RNA molecules that participate in the transcriptional and post-transcriptional regulation of gene expression through RNA interference. Pathological changes in intervertebral disc degeneration (IVDD) have been associated with alterations in the expression of microRNAs and circular RNAs, which start regulating gene expression in disc cells differently.

For example, miR-665 has been shown to decrease the expression of aggrecan and collagen type II while increasing the expression of MMP-3 and MMP-13 by inhibiting the growth differentiation factor 5 (GDF5) expression in nucleus pulposus (NP) cells.

MiR-21 can inhibit the expression of HIF-1α and VEGF in annulus fibrosus (AF) and NP cells, as well as induce apoptosis in nucleus pulposus cells. MiR-21 can promote the proliferation of NP cells and increase the expression of MMP-2 and MMP-9 mRNA.

MiR-146a inhibits the expression of IL-1-mediated inflammatory genes and catabolic proteases mRNA, as well as the levels of IL-1-mediated MMPs and aggrecanase protein. Estrogen (17β-estradiol, E2) plays an important role in inhibiting apoptosis of cartilaginous endplate cells and restoring their viability. Analysis has shown that estrogen receptor α (ERα) is a target of miR-221. It is possible that miR-221 may influence the protective effect of estrogen on degenerating cartilage endplate cells [58].

The expression levels of miR-141, miR-27a, and miR-494 are elevated in damaged NP, leading to accelerated degeneration by activating the NF-κB signaling pathway. Specifically, it has been shown that knockout of miR-141 significantly inhibits the expression of MMP-13. MiR-27a activates NF-κB signaling and significantly increases the production of pro-inflammatory factors IL-1β, IL-6, and TNF-α. MiR-494 promotes degradation of the extracellular matrix and apoptosis of degenerated human NP cells [32].

MiR-143-5p has a negative effect, inducing apoptosis and reducing the synthesis of collagen type II and aggrecan through the activation of the AMPK pathway.

MiR-640 increases in response to TNF-α and IL-1β stimulation and participates in the degenerative process by inducing senescence and apoptosis in NP cells, increasing the expression of MMP-3 and MMP-9, and reducing the levels of aggrecan and collagen type II.

circVMA21 is significantly decreased in pathological disc tissues compared to controls. Overexpression of circVMA21 restores the balance between anabolism and catabolism of the extracellular matrix in NP cells [59].

These findings highlight the role of microRNAs and circular RNAs in intervertebral disc degeneration, apoptosis, inflammation, and mechanobiology.

Guo et al. reported that the expression of circ-GRB10 was decreased, while the expression of miR-328-5p was increased in pathological intervertebral disc (IVD) samples compared to normal ones.

Increased expression of circ-GRB10 suppressed apoptosis of nucleus pulposus (NP) cells under nutrient deprivation in vitro. Further experiments showed that circ-GRB10 acts as a regulator of NP cell survival by limiting the activity of miR-328-5p and subsequently stimulating the expression of ERBB2. Inflammation was found to increase the level of circ-4099, which led to a decrease in the expression of IL-1β, TNF-α, and prostaglandin E2 in NP cells [60].

Thus, in response to cellular damage, IVD cells begin to express various RNAs that can either enhance or inhibit the pathological process. However, considering the large quantity and low tissue specificity of these molecules, it is premature to establish a cause-and-effect relationship. Currently, microRNAs can only be considered as an integral indicator of inflammatory process activity and the severity of degeneration.

### 7.2. Structural Protein Alterations

Collagen type IX is a heterotrimeric protein encoded by three genes: COL9A1, COL9A2, and COL9A3.

The presence of the Trp2 allele of the COL9A2 gene is associated with an increased risk of intervertebral disc degeneration (IVDD) in the Chinese Asian population aged 40–49. Patients with the Trp2 allele exhibited more severe degeneration of the intervertebral discs, and the association between the allele and the disease was mediated by age [61]. However, no association between this allele and the disease was found in the Japanese population [62]. Annunen also found that accelerated degeneration of intervertebral discs was associated with the Trp2 allele of the COL9A2 gene [63].

These findings indicate that allelic variations in COL9A2 associated with lumbar disc degeneration depend on both racial and ethnic characteristics.

The Trp3 allele of COL9A3 was also associated with the development of IVDD in the Finnish population in individuals with the TT genotype compared to those carrying the GG or GT genotypes. Kales et al. did not find a statistically significant association between the Trp3 allele in the Greek population of patients and the disease. In several studies [64,65], it was noted that the T allele of COLIA1 may influence gene regulation, resulting in increased expression levels and disruption of the protein formed.

Genes encoding collagen types IV and X are also associated with the disease, but there is currently insufficient data on their role. Further research is needed to establish their role in the development of IVDD [66,67].

Mio identified a strong association between the single nucleotide polymorphism (SNP) c.4603C-T in the COL11A1 gene and lumbar disc herniation in Japanese patients with lumbar disc disease [68].

### 7.3. Aggrecans

Patients with short alleles of aggrecan with repeat numbers ranging from 18 to 21 (the most common being 26–28 repeats) had the lowest amount of chondroitin sulfate chains in aggrecan molecules, which was associated with a high risk of developing intervertebral disc degeneration (IVDD) [69].

### 7.4. Vitamin D Receptor

The study [70] showed that allelic variation in the vitamin D receptor gene was associated with severe disc degeneration. Overall, the vitamin D receptor (VDR) plays an important role in normal bone mineralization and remodeling. It is not fully understood how genetic variants of the VDR gene influence intervertebral disc degeneration (IVDD). It is hypothesized that this polymorphism alters the structure of the intervertebral disc matrix [69].

### 7.5. Metalloproteinase

Takahashi found that a polymorphism in the MMP-3 gene (5A/6A) was associated with the development of intervertebral disc degeneration (IVDD) in elderly Japanese individuals (aged 64 to 94 years). The most common genotype observed in the normal population was 6A6A, while genotypes 5A5A and 5A6A were more prevalent in individuals with IVDD [71].

### 7.6. Interleukin

IL-1 increases the expression of matrix-degrading enzyme genes, such as MMP-3, MMP-9, and MMP-13. At the same time, IL-1 reduces the expression of proteoglycan and collagen II in normal disc cells. Additionally, cell culture studies using cells derived from degenerative discs have shown increased regulation of the ADAMTS-4 enzyme gene expression in the presence of IL-1. This suggests that local elevation of IL-1 levels may lead to disc dehydration and a subsequent decrease in disc height, thereby increasing the risk of developing IVDD. It has also been shown that the presence of the IL-1[alpha] allele increases the risk of developing intervertebral disc degeneration more than threefold compared to individuals without this allele [28,29,30,31,32,33,34,35,36,37,38,39,40,40,41,42,43,44,45,46,47,48,49,50,51,52,53,54,55,56,57,58,59,60,61,62,63,64,65,66,67,68].

IL-6 acts through the JAK/STAT, Ras, and PI3K16 pathways and plays an important role in the development of neuropathic pain. It has been shown that patients with pain syndrome often exhibit the IL-6 GGGA haplotype [14].

Hanaei investigated the polymorphism of IL-2 +166G/T, IL-2 -330G/T, IL-12−1188A/C, IFN-γ+847A/T, IL-4 (rs2243248 (1098G/T), rs2243250 (590 C/T), rs2070874 (33 C/T)), and 1 SNP IL-4RA (rs180275, þ1902 A/G) in patients with intervertebral disc degeneration using PCR. Among the patients, the TG IL-2−330G/T and GT IL-2 haplotypes (consisting of -330G/T and +166G/T SNPs) were prevalent, as well as the C allele of IL-4 rs2070874 and the CC genotype [72,73].

In conclusion, based on the above data, it can be proposed that polymorphisms in structural proteins and enzymes may only be considered predisposing factors in certain populations. Therefore, the role of genetics in the development of IVDD remains inconclusive at this stage. Further population studies are needed to obtain a more complete understanding of the impact of genetic factors on the development of the disease.

## 8. Discussion

Degenerative changes in the intervertebral disc (IVD) are a complex process that disrupts the cushioning function of the IVD, leads to discogenic pain, and affects the overall movement dynamics of the spinal functional segment.

Excessive mechanical load or physical disruption of disc tissues at the molecular level manifests as a series of pathological processes. Diffuse nutrition of IVD cells is disrupted, and there is active degradation of the extracellular matrix by metalloproteinases and ADAMTS. Nucleus pulposus (NP) and annulus fibrosus (AF) cells, in response to changes in the extracellular matrix, alter gene expression and start producing cytokines that promote vascular ingrowth and nociceptive receptor formation within the IVD.

Cytokine expression, including TNF-a, IL-6, IL-8, PGE2, COX, MMP3, and MMP7, is enhanced by infiltrating immune cells in the IVD. These immune cells are not destroyed by FasL, as its expression in NP cells decreases due to gene expression changes. Immune cells recognize the damaged disc as an antigen and also begin to secrete pro-inflammatory mediators, creating a “cytokine storm” in the IVD tissue. This process promotes the differentiation of B lymphocytes into plasma cells, which secrete IgG antibodies against collagen types II and V, as well as aggrecan. The presence of antibodies against structural components of the IVD initiates a slow process of degenerative changes in the disc [74].

An important role of low-grade inflammation in IVDD could be as a factor predisposing to multilevel degeneration in high-risk individuals, such as the elderly and patients with metabolic syndrome. Excess adipose tissue and insufficient immune system activation can lead to chronic inflammation all over the body, resulting in impaired tissue function, particularly in IVDs [25]. Identification of patients with manifestations of low-grade inflammation could help to adjust treatment strategies and delay or even prevent disease onset.

The presence of genetic mutations and polymorphisms in the genes for collagen I, IX, XI and aggrecan significantly increases the risk of developing IVDD. Mutations in this group of genes lead to changes in the mechanical properties of the IVD and reduce its ability to resist injury. However, these factors only indicate the risk of the development of IVDD, but not its manifestation.

The status of the intervertebral disc (IVD) as an immune-privileged structure remains a subject of debate: whether only the IVD is considered an immune-privileged organ or if other cartilaginous tissues in the body can also be classified as privileged structures.

The immune privilege of certain organ structures can be explained by the characteristics of embryological development. NP (nucleus pulposus) cells of the IVD originate from the notochord, which is formed during the third week of embryonic development. By the sixth and seventh week, chordal cells migrate to the central parts of the sclerotome and then differentiate into immature NP cells by the tenth week. The main phenotypic markers of NP cells include HIF1/2α, GLUT1, KRT 18/19, CA-3/12, CD24, and A2M. During early development, these cells are isolated from the immune system and surrounded by a dense fibrous ring. Diffuse nutrition is provided by blood vessels penetrating the vertebral endplate, after which they are eliminated, and in the adult state, the NP becomes an avascular structure. Blood vessels and nerves only reach the outer layers of the annulus fibrosus (AF), from where nutrient diffusion into the extracellular matrix occurs in adulthood, which accounts for the low regenerative capacity of the NP.

When comparing NP data with meniscus and articular cartilage, not only biochemical but also physiological differences are observed. The meniscus and articular cartilage are formed through the condensation of mesenchymal cells in the intermediate layer known as interzone cells.

### 8.1. Meniscus

The meniscus is partially supplied with blood in its peripheral part by branches of the genicular artery, which contributes to its ability to regenerate. The remaining portion of the meniscus receives nutrients through diffusion. The main phenotypic markers of the meniscus include C1QR, CA12, COL1A1, COL1A2, ESTs, FLJ20831, HPCAL1, LIMK2, and PDLIM1. The meniscus is primarily composed of type I collagen, with type II collagen predominating in the inner third. Avascularity is maintained through the secretion of anti-angiogenic factors, such as endostatin/collagen XVIII. Innervation of the meniscus is provided by mechanoreceptors, including Ruffini corpuscles, Pacinian corpuscles, and Golgi tendon organs [75].

### 8.2. Articular Cartilage

Articular cartilage consists of four zones: superficial, transitional (middle), deep (radial), and calcified. The main component of cartilage is type II collagen. Phenotypic markers of articular cartilage include COMP, CYTL1, FBLN1, GDF10, HIF-1/2α, IBSP, and MGP. Avascularity of articular cartilage is maintained by thrombospondin-1 (TSP1), chondromodulin-I (ChM-I), endostatin/collagen XVIII, osteonectin, and the N-terminal propeptide derived from type II collagen (PIIBNP). Therefore, the nourishment of cartilage occurs solely through diffusion, unlike the meniscus [75].

The question remains open as to whether cartilages are immune-privileged structures or if this property is specific only to the intervertebral disc (IVD). It was previously believed that avascularity and the dense extracellular matrix (ECM) of cartilage contribute to its immune privilege, resulting in a diminished ability of the immune system to attack transplants. However, this viewpoint has been challenged by several researchers who have shown that chondrocytes themselves possess antigenic properties and can elicit an immune response when interacting with immune cells. In their intact state, chondrocytes are “shielded” from the immune response due to the characteristics of the ECM. However, in cases where the matrix composition is altered, such as during transplantation or injury, immune cells, particularly NK cells and CD4 Th lymphocytes, infiltrate the area and can induce rejection or degenerative processes, especially in the case of transplantation [76].

Another insufficiently studied aspect of the development of degenerative changes in the intervertebral disc (IVD) is the question of the primary event that leads to the onset of the disease.

Physical differences between individuals, such as disc height, thickness of the endplates, speed of inflammatory response, intensity of the inflammatory cascade, rate of vascular and nerve ingrowth, characteristics of collagen structure, and the ratio of collagen to aggrecan, as well as different types of collagen, can serve as predictors of mechanical tissue rupture. Exploring these patterns could help answer the question of which specific mechanism of intervention to choose in order to halt the progression of degenerative disc disease (DDD).

The first event in the pathogenesis of IVDD is a mechanical impact on the disc, which leads to a violation of the integrity of the fibrous ring (formation of a crack) and prolapse of the NP, which then has subsequent effects on nearby nervous structures. Thus, in individuals with genetic mutations in structural proteins, fibrous ring density may not be sufficient to counteract even minor trauma.

It is possible that the gene expression by NP cells and the production of angiogenesis and innervation factors in the disc are attempts at regenerating the structures of the intervertebral disc (IVD). This leads to an improvement in cell nutrition through enhanced blood flow to the disc. However, due to the unique embryological development of the IVD, it is recognized by the immune system as a foreign antigen, triggering an autoimmune process and exacerbating degenerative processes.

Some studies have also found a correlation between the inflammation activity and the size of the extruded fragment [77], which can be explained by the secondary traumatization of the fibrous ring due to an inflammatory reaction. However, it is unclear how this observation is consistent with the phenomenon of hernia resorption.

Another question arises: can a deficiency in nutrient diffusion into the disc be a primary cause of intervertebral disc degeneration (IVDD), leading to a decrease in its elastic properties, changes in gene expression, and disc protrusion? Or is excessive mechanical loading the primary cause? In such a case, another question arises: will treatment strategies differ depending on the primary source of disease development?

Therefore, immune reaction also plays a positive role in IVD recovery. In view of this, an issue comes up regarding to what extent it is possible to modulate the immune response.

The exact primary cause of IVDD is still a subject of debate. Both insufficient nutrient diffusion and excessive mechanical loading have been proposed as potential factors contributing to disc degeneration. If nutrient diffusion deficiency is identified as the primary cause, strategies aimed at improving nutrient supply to the disc may be considered in the treatment approach. On the other hand, if excessive mechanical loading is determined to be the primary cause, interventions focused on reducing mechanical stress on the disc may be more appropriate. Further research is needed to better understand the underlying mechanisms and to develop targeted treatment strategies based on the primary source of disease development.

Summarizing the above, it can be concluded that the initial event in the development of IVDD (intervertebral disc degeneration) is a primary injury, during which micro- or macroscopic cracks in the fibrous ring are formed. Factors predisposing to such damage include the overall functional status of the body, the condition of the musculoskeletal system, and the presence of systemic diseases, including diabetes, metabolic syndrome, and certain genetic and rheumatological conditions. In addition, the strength of the disc–vertebral complex is determined by the presence of single nucleotide polymorphisms (SNPs) and certain alleles in structural proteins such as collagen and aggrecan. Therefore, carriers of corresponding genetic variants are less able to withstand external damaging influences.

After the formation of cracks, a cascade of events is initiated, which can be characterized as a balance between reparative and destructive forces. Inflammation is one of the leading factors in disc degeneration and destruction, and its intensity is partly determined by individual genetic characteristics. In the case of a predominance of reparative processes, a cascade of “vicious” reactions that support inflammation is not formed, and the disc is restored and continues to perform its physiological functions. Conversely, a local inflammatory focus is formed, leading to neovascularization, neurogenesis, and possibly triggering a cascade of autoimmune reactions. Over time, these factors contribute to the progression of disc degeneration and the clinical manifestations of the disease (Figure 1).

## 9. Conclusions

The intervertebral disc (IVD) is a complex functional element of the spine that significantly differs from other cartilages. The functioning of the IVD and its ability to withstand axial load imply a complex macroscopic, microscopic, and molecular structure. Disc degeneration is a complex, multifactorial process consisting of factors that can be classified as necessary, sufficient, or background factors. Thus, we hypothesize that the primary factor for the development of intervertebral disc degeneration (IVDD) is the formation of a “crack” or initial injury, which can be both a sufficient and an insufficient factor depending on the circumstances. Concurrent background conditions, including genetic factors and immune reactivity, may contribute to the progression or development of the disease. Despite the wealth of accumulated information, larger prospective studies are needed to support this viewpoint and identify significant early-disease biomarkers.

## Figures and Tables

**Figure 1 biomedicines-11-02714-f001:**
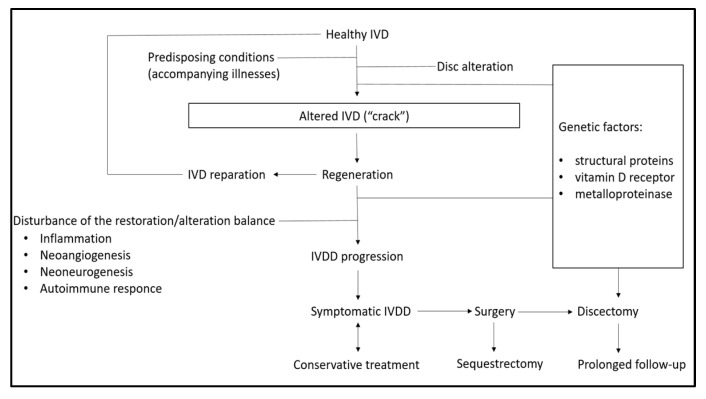
Pathogenesis of intervertebral disc disease. IVD–intervertebral disc; IVDD-intervertebral disc disease.

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
