# Peer review of "Understanding Intervertebral Disc Degeneration: Background Factors and the Role of Initial Injury"

_biomedicines, 2023, doi:10.3390/biomedicines11102714_

Round 1

Reviewer 1 Report

Nice article but the title is broad, so not all disc conditions are discussed in the article so authors need to change title to reflect content

IMportant articles are not cited as:

Zhao L, Manchikanti L, Kaye AD, Abd-Elsayed A. Treatment of Discogenic Low Back Pain: Current Treatment Strategies and Future Options-a Literature Review. Curr Pain Headache Rep. 2019 Nov 9;23(11):86. doi: 10.1007/s11916-019-0821-x. PMID: 31707499.

Minor editing

Author Response

Thank you for the thorough review. We are happy to take all your comments into consideration.

Please find attached revised version of the manuscript with highlighted changes.

Here is a point by point answer on Your comments:

Nice article but the title is broad, so not all disc conditions are discussed in the article so authors need to change title to reflect content

We have changed the article heading to “Understanding Intervertebral Disc Degeneration: Background factors and the role of initial injury”. We believe that this heading better reflects the contents of the article.

Important articles are not cited as:

Zhao L, Manchikanti L, Kaye AD, Abd-Elsayed A. Treatment of Discogenic Low Back Pain: Current Treatment Strategies and Future Options-a Literature Review. Curr Pain Headache Rep. 2019 Nov 9;23(11):86. doi: 10.1007/s11916-019-0821-x. PMID: 31707499.

We have included the above reference in the text (# 2 on the “Reference” section, p.2).

Reviewer 2 Report

This review by the Drs. Karchevskaya and colleagues encompasses the latest work in the field of a very interesting topic - the etiopathology of the intervertebral disc degeneration (IVDD). It aims to improve the understanding of the latest findings and propose possible mechanisms to avoid surgical interventions. This review has been very well designed, executed and was very neatly written. Moreover, it is potentially very interesting to a broad audience of readers. Overall, it has been a pleasure to read and I have little to nothing to add.

Good quality of English.

Author Response

Thank you for the thorough review and high rating of our work. Due to the comments from other reviewer we decided to change the article heading to “Understanding Intervertebral Disc Degeneration: Background factors and the role of initial injury”. We believe that this heading better reflects the contents of the article.

Kind regards,

Yuri Poluektov

Reviewer 3 Report

The article is scientifically sound and explains the necessity of studying different aspects of IVVD can be much more explained. Conclusion should be elaborated with the detailing aspects of the study 

The quality of English is good and explained in a simple manner so that non scientific readers can also access

Author Response

Thank you for the thorough review. We are happy to take all your comments into consideration. 

Here is a point by point answer on Your comments:

The article is scientifically sound and explains the necessity of studying different aspects of IVVD can be much more explained. Conclusion should be elaborated with the detailing aspects of the study 

We have rewritten the conclusion of the MS to make it more sound and clear (p. 14).

The quality of English is good and explained in a simple manner so that non scientific readers can also access

We have revised the text of the MS.

Kind regards,
Yuri Poluektov